# Intracardiac Echocardiogram: Feasibility, Efficacy, and Safety for Guidance of Transcatheter Multiple Atrial Septal Defects Closure

**DOI:** 10.3390/jcm11092394

**Published:** 2022-04-24

**Authors:** Jae-hee Seol, Ah-young Kim, Se-yong Jung, Jae-young Choi, Yeon-jae Park, Jo-won Jung

**Affiliations:** 1Division of Pediatric Cardiology, Department of Pediatrics, Congenital Heart Disease Center, Severance Cardiovascular Hospital, Yonsei University College of Medicine, Seoul 03722, Korea; seoljh0623@gmail.com (J.-h.S.); dalkiay@yuhs.ac (A.-y.K.); jung811111@yuhs.ac (S.-y.J.); cjy0122@yuhs.ac (J.-y.C.); 2Department of Pediatrics, Yonsei University Wonju College of Medicine, Wonju 26493, Korea; 3Department of Biostatistics, Yonsei University Wonju College of Medicine, Wonju 26493, Korea; aceail@yonsei.ac.kr

**Keywords:** real-time three-dimensional transesophageal echocardiography, intracardiac echocardiography, multiple atrial septal defects, transcatheter closure

## Abstract

We aimed to determine the feasibility, efficacy, success, and safety of intracardiac echocardiography (ICE) in transcatheter multiple atrial septal defect (ASD) closure. Of 185 patients with multiple ASDs who underwent transcatheter closure, 140 (76%) patients who weighed <30kg with a narrow distance between defects or in whom single device closure was anticipated were guided by ICE and 45 patients were guided by three-dimensional (3D) transesophageal echocardiography (TEE) with or without ICE. Patients in the ICE group were relatively younger and weighed less than those in the 3D TEE group (*p* < 0.0001). The ratio of the distance between defects >7 mm was high, and more cases required ≥2 devices in the 3D TEE group than those in the ICE group (*p* < 0.0001). All patients in the 3D TEE group and seven patients (5%) in the ICE group were operated on under general anesthesia (*p* < 0.0001). The fluoroscopic time was shorter in the ICE group (13.98 ± 6.24 min vs. 24.86 ± 16.47 min, *p =* 0.0005). No difference in the complete closure rate and complications was observed. ICE-guided transcatheter and 3D TEE were feasible, safe, and effective in successful multiple ASD device closures, especially for young children and patients at high risk under general anesthesia.

## 1. Introduction

Transcatheter atrial septal defect (ASD) closure has become the treatment of choice as a minimally invasive approach that allows for rapid recovery and earlier hospital discharge. This procedure, guided by proper echocardiographic imaging, is becoming the gold standard treatment [1,2]. Preoperative and intraoperative echocardiographic imaging provides information on the number of defects, morphology, sizes, and surrounding rims to make treatment decisions. Cases of multiple ASDs account for 4–10% of the various types of ASDs [3]; however, transcatheter ASD closure using multiple devices is challenging in terms of device selection and positioning. Transcatheter closure of multiple ASDs should consider the complex defect morphology and higher risk of complications, including device embolization, residual shunt, and device erosion. Therefore, acquiring an excellent intraprocedural echocardiographic image is essential for a successful procedure. We can choose the appropriate devices with comprehensive ASD imaging and can reduce the fluoroscopy exposure time [4,5].

Two-dimensional (2D) echocardiography, specifically transesophageal echocardiography (TEE) with or without ICE, has conventionally been used for transcatheter ASD closure. However, three-dimensional (3D) TEE plays a central role, especially in complex cases with multiple ASDs [6,7,8,9,10]. In particular, this procedure provides a superior evaluation of the anatomic structure of ASDs by visualizing an en face view of the defect size, shape, residual rim, and spatial relationship between the defects and helps guide their closure [11,12]. However, 3D TEE is usually performed with adult-sized probes to obtain acceptable results, which can be safely used in children who weigh at least >30 kg [13]. Using 3D TEE also requires general anesthesia, limiting its use [14,15]. In young children and a few adult patients at a high risk when operated under general anesthesia, ICE can be selectively utilized for transcatheter closure. Many studies reported the successful use of ICE and 3D TEE in ASD device closure, and they agreed on the preferential use of 3D transesophageal echocardiography. However, there is a lack of reported data demonstrating the comparable effectiveness of ICE and 3D TEE in multiple atrial defect closure. Additionally, there is limited knowledge regarding the effectiveness of ICE in accurately describing the size and characteristics of multiple ASDs and/or positioning of multiple devices. The usefulness of 3D TEE in multiple ASD device closures has already been confirmed through many studies. Here, we wanted to assess through retrospective observation the feasibility, efficacy, procedural success, and safety of ICE in transcatheter multiple ASD closure by comparison with 3D TEE.

## 2. Materials and Methods

### 2.1. Study Protocol and Population

Overall, 185 patients with multiple ASDs who underwent transcatheter device closure between January 2010 and August 2020 were enrolled in the study. All cases with more than single interatrial communication or multi-fenestrated atrial septal defects were included in the multiple ASD. We excluded patients who were referred to surgery before undergoing the procedure or with contraindications to ASD closure because of severe pulmonary hypertension. The following data were collected from the medical records: demographic data, number of defects, defect size, number of devices, total fluoroscopy time, procedural complications, and clinical outcomes. Before performing the procedure, information on the number of defects and adequacy of the septal rims for the device was obtained from TEE or cardiac computed tomography, and 2D ICE or 3D TEE was performed before the procedure based on the defect characteristics and patient factors, including age, body weight, the risk to general anesthesia, comorbidities, and overall general condition. In particular, we preferred using ICE, predominantly in small children weighing <30 kg regardless of age, or with a closer distance between the multiple defects in the pre-procedure transthoracic echocardiography. This procedural strategy was primarily based on a recommendation regarding multiple ASD device closure, which stated that multiple ASDs with an inter-defect distance of <7 mm can be successfully closed using a single device [3,16,17]. Some patients required both 3D TEE and ICE; however, 3D TEE was the primary imaging device used in these cases. We compared the procedural data and outcomes between the 2D ICE and 3D TEE (±ICE) groups.

According to the pre-procedural and intraprocedural assessments, we classified atrial septal anatomy into aneurysmal type, fenestration type, and septal aneurysm with fenestration type and classified whether there are fenestrations. If it did not correspond to aneurysm or fenestration, it was classified as “neither”. The fenestrated ASD was defined as an ASD with small, multiple fenestrations or pinpoint defects. According to Silver et al. [18] and Pearson et al. [19], an atrial septal aneurysm was defined as the protrusion of an aneurysm >10 mm beyond the plane of the atrial septum into either the right or left atria.

Procedural success was defined as a stable, adequately positioned device in the absence of trauma to the surrounding structures. The medical report in the pointed data registry demonstrating residual defects, major and minor complications, including cardiac tamponade, erosion, device embolization, thromboembolism, and complete atrioventricular block, and the need for re-intervention were included in the follow-up outcomes. The study endpoint was the evaluation of the difference in the procedural success and complications between the 2D ICE and 3D TEE (±ICE) groups.

We defined “feasibility” and “safety” of using ICE as the cases where closure was successfully complete without any significant residual shunt with no minor and major complications.

### 2.2. Procedural Technique

Echocardiographic imaging was performed to assess the defect diameter, intra-defect distance, superior, inferior, and anterior rims of the defect, total antral septal length, the proximity of the defect to surrounding structures, and final device positioning. Our institution performed ASD closure with TEE guidance under general anesthesia until mid-2011. Since 2011, ICE has been used as the standard echocardiographic imaging during transcatheter closure of ASDs and patent foramen ovales in the pediatric and adult populations at our institution. TEE was rarely performed, except for the cardiac catheterization laboratory when 3D TEE was required. Procedures in pediatric patients guided by ICE were performed under local anesthesia with mild sedation. Patients with a minor distance between two significant defects on pre-procedural TEE and those in whom one device was expected to close another defect underwent transcatheter closure with 2D ICE. Otherwise, 3D echocardiography was used.

Furthermore, ICE was used when a patient’s body weighed <30 kg. We followed the ICE -guided ASD closure methodology previously described by Earing et al. [20]. ICE was performed with the ACUSON AcuNavTM 8-Fr ultrasound catheter (ACUSON, Issaquah, WA, USA) and an 8- or 8.5-French short-sheath catheter introducer, and 3D TEE was performed using the Philips EPIQ 7c (Philips Medical System Andover, MA, USA) and probe. The defect sizes were measured using color echocardiogram or balloon sizing. Balloon sizing was used in selected cases depending on the defect size, atrial septal wall characteristics, and operator preference.

The ASDs in our patients were closed using the Amplatzer septal occluder (Abbott, Chicago, IL, USA), Cocoon Septal occluder (Vascular Innovations Co Ltd., Nonthaburi, Thailand), Occlutech Figulla Flex II device (Occlutech, Helsingborg, Sweden), and Gore Cardioform Septal Occluder (Gore Medical, Newark, DL, USA). When more than one device was required, the operator ensured that all devices were positioned at adequate distances from important structures, such as the vena cava entrances and coronary sinus, because the devices may interfere with blood flow and increase the risk of thrombosis. TEE was performed immediately after each procedure to confirm the position and stability of the device(s) and residual shunt. All patients were evaluated using transesophageal echocardiography, electrocardiogram, and chest radiograph on the day of and on the day after the procedure. Patients were followed-up after 1 week; at 3, 6, and 12 months; and annually thereafter until the first documented event or time of final follow-up, whichever came first.

### 2.3. Statistical Analyses

All analyses were performed using SPSS version 25.0 (IBM Corp., Armonk, NY, USA). The results were reported as numbers (percentage), mean ± standard deviation, and range. Comparisons of the baseline data, procedure-related characteristics, and outcomes between the 2D echocardiography and 3D TEE ± ICE groups were performed. Fisher’s exact test was performed for categorical variables, whereas the independent *t*-test was performed for continuous variables. A *p*-value of ≤0.05 was considered statistically significant.

### 2.4. Ethics Statement

This study was approved by the Yonsei University College of Medicine Institutional Review Board and Research Ethics Committee of Severance Hospital (study approval number: 2021-2324-001). The requirement for individual consent was waived because of the retrospective study design.

## 3. Results

In total, 185 patients with multiple ASDs underwent transcatheter device closure at our institution during the study period. Table 1 shows the demographic and pre-procedural characteristics of these patients. We examined 56 male and 129 female patients with a mean age of 22 years (range, 9 months–77 years) and mean weight of 36.5 kg (range, 5–94.9 kg). A total of 140 (76%) and 45 (24%) patients underwent transcatheter device closure under 2D ICE and 3D TEE (±ICE) guidance, respectively. Thirteen patients (28.9%) in the 3D TEE group also underwent ICE. The mean age of the 2D ICE group was 15.1 years, which was younger than that of the 3D TEE group, which was 43.8 years (*p* < 0.0001). The mean bodyweight of the 2D ICE group was 28.7 kg (range, 5–94.4 kg), which was less than that of the 3D TEE group (*p* < 0.0001). The 2D ICE and 3D TEE (±ICE) groups most commonly had two defects; however, the proportion of patients with more than three defects was significantly higher in the 3D TEE group (*p* = 0.02).

The procedural data of both groups are demonstrated in Table 2. The septal type without aneurysm or fenestration was the most common (56%), followed by the presence of an aneurysm (28%). No significant difference in septal type between the two groups was observed. In the 2D ICE group, 103 of 140 (73%) patients had a distance between defects less than 7 mm, and in the 3D TEE group, 17 of 45 (38%) patients had a distance of less than 7 mm and the rest had more than 7 mm. There were many patients with a relatively short distance between defects in the ICE group, and in the group using 3D TEE, there were many patients with a distance between defects of more than 7 mm. In the 3D TEE group, the proportion of patients who required two or more devices was higher than that in the 2D ICE group (*p* < 0.0001), and 17 (40%), 22 (49%), and 5 (11%) patients required one, two, and three devices, respectively. The fluoroscopy time was significantly shorter in the 2D ICE group (*p* = 0.005); however, this may be because a majority of the patients in the 2D ICE group only required one device or had simple defects. All patients in the 3D TEE group were operated under general anesthesia.

Among the 185 patients who underwent ASD closure, 181 (98%) cases were successful (Table 3). Three cases were converted to open surgery, which included one patient from the 2D ICE group who demonstrated undesirable device interference and two patients from the 3D TEE group who were detected with device migration into the left ventricle and complete atrioventricular block, respectively. Immediate post-procedural complications were observed in one patient who received two devices under 3D TEE guidance; one of the two devices had embolized into the left atrium a few hours after the procedure. The patient subsequently underwent successful transcatheter retrieval of the embolized device. The remaining defects in this patient were monitored on an outpatient basis, and the patient did not require additional device closure. Patients with successful closure were monitored conservatively over a mean duration of 49.7 (range, 1–126) months. In the immediate post-procedure period, residual primary defects were noted in 129 (93%) and 40 (89%) patients in the 2D ICE and 3D TEE groups, respectively. At the final follow-up, only 23 (12%) and 12 (27%) patients in the 2D ICE and 3D TEE groups, respectively, demonstrated significant residual primary defects. Other isolated defects were noted in 34 (24%) and 8 (18%) patients in the 2D ICE and 3D TEE groups, respectively; however, these were not significant and did not require re-intervention. Among the patients with residual defects, 17 were lost to follow-up. Complete closure was confirmed using TEE in 88 (63%) and 28 (62%) patients in the 2D ICE and 3D TEE groups, respectively. No significant difference in the outcomes of the groups was observed. No severe long-term complications, including erosion, device embolization, thromboembolism, and atrioventricular block, were observed. Furthermore, procedure-related mortality was not observed during follow-up.

In all children weighing less than 10 kg, device closure was performed safely and successfully with ICE. The youngest child was 9 months old and the smallest weighed 5 kg; 79% of the children had an inter-defect distance of less than 7 mm. All of these patients weighing less, including those with an inter-defect distance of larger than 7 mm, were treated using one device for ASD closure. For this reason, either a residual shunt or an isolated another defect was observed immediately after the procedure. However, at the latest follow up, 75% of patients had complete closure. These results are similar to those of all other patients (Appendix A). Mobitz type II AV block occurred briefly in one child during the procedure, but improved device closure was successfully completed. There were no other arrhythmias that occurred during follow-up. For two days after the procedure, the presence of hematoma around the treated vessel and the occurrence of arrhythmia were observed, and all were discharged without any abnormal findings. No patient in this series developed atrial arrhythmias as a post procedural complication of ICE use.

## 4. Discussion

Transcatheter ASD closure under echocardiographic guidance is the gold standard treatment for most cases of ASDs, including complex cases with insufficient rims or multiple ASDs [1,21]. In the 2019 guidelines for TEE and 2015 guidelines of echocardiographic assessment for ASD closure from the American Society of Echocardiography, there is insufficient evidence for the standard imaging guidance of multiple ASD device closure [13,22]. The primary challenge in patients with transcatheter closure of multiple ASDs is that the device may be hard to position properly, which increases the risk of complications, including device embolization, cardiac erosion, thrombus formation, and arrhythmia [5]. Understanding and visualizing the anatomy of the defects, including the surrounding and intervening rims during the procedure, is the cornerstone of successful multiple ASD closure and potentially improves the efficiency and safety profiles of the procedure. In this respect, both 3D TEE and ICE provide good-quality images that increase the chances of successful cardiac catheterization. The novelty of this study is that it demonstrated successful transcatheter closure of multiple ASDs under the guidance of 3D TEE with the predominant usage of 2D ICE alone in selected patients.

Notably, 3D TEE is an excellent procedure for patients with ASDs of complex anatomies, such as multiple defects, because it provides a high spatial resolution of all defects, implanted devices, and any residual shunting [11,23,24,25,26,27,28]. Several studies have demonstrated that 3D TEE adequately visualizes the interatrial septum of complex ASDs during transcatheter ASD closure [29,30,31,32]. In our study, 3D TEE was successfully performed to determine the rims and relationships between defects. The 3D TEE group in this study was more likely to receive two or more devices than the 2D ICE group. If the distance between the significant size of defects requiring closure is >7 mm, we selected the 3D TEE more often than 2D ICE; 3D TEE facilitated the successful placement of up to three devices during transcatheter closure (Figure 1). This suggests excellent utility of 3D TEE in complex defects.

On the contrary, the major limitation of 3D TEE is that it cannot visualize the entire defect in a single window, principally when the posteroinferior rim of the defect is blocked by the inferior vena cava, limiting the visualization while passing the wire and catheter during or after device implantation (Figure 2). Furthermore, 3D requires general anesthesia, and the sizeable 3D TEE probe cannot be used in children who weigh <30 kg. Taniguchi et al. [33] performed transcatheter ASD closure under 3D TEE guidance in a 7-year-old patient. In our study, the youngest patient who underwent closure under 3D TEE guidance was 10 years old and weighed 46 kg. The lowest weight in our study was 44.4 kg.

Compared with 3D transesophageal echocardiography, ICE provides relatively clearer images of the posterior and inferior walls of the atrial septum [4,20,34,35]. There are several studies that reported safe and effective ASD device closure using ICE in patients including children [8,34,35,36]. Our institution has utilized ICE in solitary ASD closure for children and adult patients regardless of their weight since late 2011. The lightest patient who underwent multiple ASD closure under ICE guidance in our study weighed 5 kg. ICE provided excellent images of the defects’ size, location, and relationship with the surrounding tissues. ICE may occasionally cause atrial arrhythmia or vessel injury especially in small children during procedure; however, it has the distinct advantage of not requiring general anesthesia. The ICE probe is also easily manipulated by a single operator, allowing the operator complete control over image acquisition. This can result in reducing fluoroscopic time and procedure time. Not requiring general anesthesia is an important advantage of ICE compared with TEE, especially in people who are older. ICE has a disadvantage in cost compared with TEE, but it can be overcome when considering that ICE can reduce the cost of general anesthesia and anesthesia personnel.

Despite its advantages, there are limited data on the usefulness of ICE in multiple ASD closure. In our study, 2D ICE was used for patients unable to use 3D TEE due to low body weight or risk of general anesthesia and for patients with relatively less complex defects that do not require multiple devices. We performed transcatheter closure under 2D ICE guidance in 76% of patients with multiple ASDs. Under ICE guidance, multiple ASD device closure was successful in all patients using ICE and 11 patients were able to receive two devices without complication including interference between devices and device embolization (Figure 3). The lightest patient who underwent multiple ASD closure under ICE guidance in our study weighed 5 kg. The youngest patient who received two devices with ice was 1 year and 9 months old and weighed 10.9 kg. Most patients in the 2D ICE group (*n* = 129, 92%) needed only one device because a narrow enough inter-defect distance device disc can close multiple defects and/or other isolated defects were not significant enough to require additional device closure (Figure 4).

During the follow-up period, there was no significant complication, including arrhythmia, embolization of device, or thrombosis in either groups. At the latest follow-up, the overall complete closure rate was 63.7% There was no difference in the complete closure rates between the two groups. All residual primary defects and other isolated defects were not significant shunt requiring additional device closure. In the 3D TEE group, there were many more complicated cases of multiple ASDs, but in the 2D ICE group, it is meaningful in that there were more children with younger age (mean ± SD, 15.1 ± 20.3 (9 months–77 years) vs. 43.9 ± 18.8 (10 months–76 years)) and small body weight (mean ± SD, 28.7 ± 23.3 (5–94.4 kg) vs. 60.9 ± 11.0 (44.4–94.9 kg)). Children under 10 kg treated with ICE showed similar results compared with all children. In particular, it suggests the feasibility and safety of ICE in multiple ASDs in that even children under 10 kg treated with ICE showed similar outcomes compared with all other patients.

Because the shape of the defects can be elliptical or complex ovoid, and asymmetric shape, the more complex the defect, the greater the difference in size in 2D ICE and 3D TEE. A study comparing ASD dimensions using 3D TEE vs. 2D TEE and ICE showed that 2D TEE and ICE are more likely to provide a smaller long-axis dimension and larger short-axis dimension when compared with RT3D TEE [6]. The difference in measurement between the 2D echo and 3D TEE is insignificant, and it is unclear how it affects the device selection. To overcome this limitation, we tried to select the optimal device by referring to the balloon dimension based on the stop flow technique. In this study, there were only 13 cases where 3D tee and ICE were used simultaneously, so it is difficult to directly compare the measurement of ASD by 3D TEE and ICE. In further studies, we can study how this affects device selection through comparative evaluation.

Since each echocardiographic modality has advantages and disadvantages in multiple ASD device closure, the stability and efficiency of the procedure can be improved by selecting each modality in consideration of the characteristics of the defect and the patient’s condition; 3D TEE is particularly useful for ASDs with complex geometries because it provides the complex morphology and spatial relationship between multiple defects. However, ICE can be considered the first choice in a case requiring a single device, one large defect with one or more small defects and an inter-defect distance less than 7 mm, or one large defect with a distance of less than 7 mm from the farthest point of another defect. If the distance between one large defect and another is greater than 7 mm but these multiple defects are not of significant size, ICE could be considered in patients with low body weight and those at risk of general anesthesia.

Our study has some limitations. First, due to its retrospective design, the patients were not evenly distributed between the groups. Therefore, it is difficult to directly compare ICE and 3D TEE in this study. More patients with multiple ASDs are required, and long-term complications are needed to be observed to determine the transcatheter closure experience. Second, the 3D TEE with and without ICE groups could not be compared because of the small number of patients. However, our study was meaningful, in that, it demonstrated that ICE was a reliable method for closing multiple ASDs, particularly in children and in those with simple multiple ASDs where multiple devices are not required. According to our institution’s standard practice strategy, we were able to maximize the advantages of ICE and to provide procedural guidance for multiple ASD closure through tailored decisions for individual patients.

## 5. Conclusions

In conclusion, transcatheter multiple ASD device closure using 2D ICE or 3D TEE was successfully performed without fatal complications. Both modalities can provide good images for successful transcatheter closure of multiple ASDs. Although the benefits of 3D TEE are obvious, 3D TEE may not be necessary in all cases with multiple defects. 2D ICE can be safely and efficiently performed even in multiple ASD device closure as an alternative to 3D TEE. ICE can be considered first in relatively simple ASD cases where the distance between defects is short or two or more devices are not expected to be needed, especially for patients who weigh less and those at risk of general anesthesia. There is a need for improvement in the cost aspect of ICE, and the design and function of echocardiographic tools need to provide more successful and efficient results for these interventions in the future. Furthermore, there are existing technologies that can implement 3D images using ICE. However, there are not many clinical trials yet, and there is a problem with the cost of the catheter as well. Overcoming these limitations, improving the imaging capabilities, and mastering the manipulation techniques could lead to good echocardiographic modality in cardiac interventions in patients with complex defects. As a new technology, attempts to apply EchoNavigator [37], a new technology that combines echocardiography and real-time X-rays, to transcatheter closure in children or complex multiple ASDs should also be noted.

## Figures and Tables

**Figure 1 jcm-11-02394-f001:**
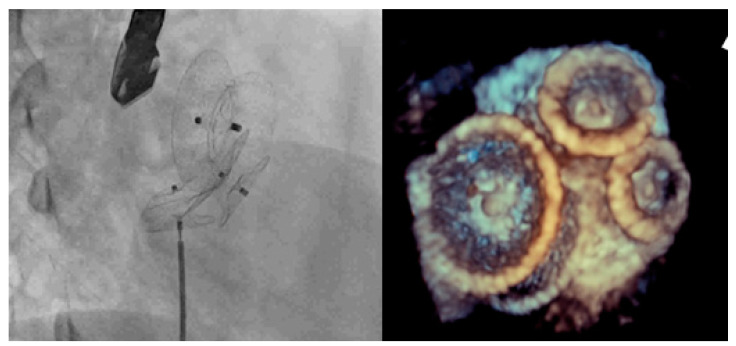
Implantation of 28, 16 and 12 mm Amplatzer septal occluders guided by real-time three-dimensional transesophageal echocardiography.

**Figure 2 jcm-11-02394-f002:**
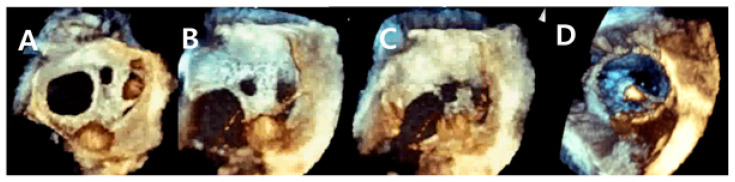
Real-time three-dimensional transesophageal echocardiography images during transcatheter closure of multiple atrial septal defects. (**A**) The defect is seen here, and its size, shape, and distance from the surrounding structures are clearly noted. The posterior and inferior septal rims are poorly visualized. (**B**,**C**) The guide catheter is visible across the atrial septum. (**D**) This photograph shows left disk expansion and stable positioning of the three septal defect closure devices.

**Figure 3 jcm-11-02394-f003:**
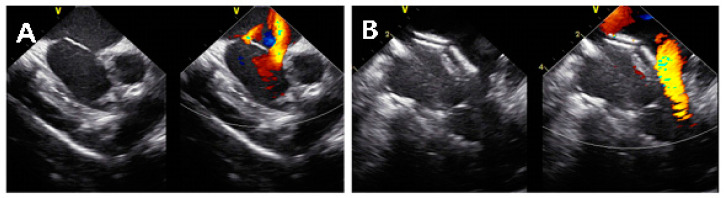
Images from intracardiac echocardiography during transcatheter closure of multiple atrial septal defects. (**A**) Color Doppler imaging shows two atrial septal defects near the posterior and aortic rims. (**B**) Two devices are seen placed within the atrial septum.

**Figure 4 jcm-11-02394-f004:**
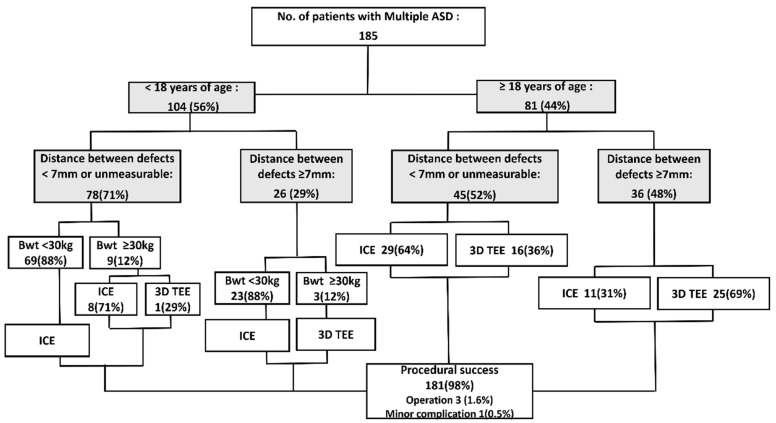
Summary of the echocardiographic tool selection strategy for patients with multiple atrial septal defects requiring transcatheter closure.ASD, atrial septal defect; Bwt, body weight; ICE, intracardiac echocardiography; RT 3D TTE, Real-time three-dimensional transesophageal echocardiography; TEE, transesophageal echocardiography.

**Table 1 jcm-11-02394-t001:** Patients’ baseline and pre-procedural characteristics.

Variables	Total(%)	ICE(%)	3D TEE ± ICE(%)	*p-*Value
	*n* = 185 (100)	*n* = 140 (76)	*n* = 45 (24)	
Age, year				
Mean ± SD	22.1 ± 23.4	15.1 ± 20.3	43.9 ± 18.8	<0.0001
(range)	(9 months–77 years)	(9 months–77 years)	(10 months–76 years)	
Sex (%)				0.8878
Male	56 (30.3)	42 (22.7)	14 (7.6)	
Female	129 (69.7)	98 (53.0)	31 (16.8)	
Bodyweight, kg	36.6 ± 25.1	28.7 ± 23.3	60.9 ± 11.0	<0.0001
Mean ± SD	(5–94.9)	(5–94.4)	(44.4–94.9)	
(range)				
ASD defect count (%)				0.0231
2	118 (63.8)	95/140 (67.9)	23/45 (51.1)	
3	39 (21.1)	23/140 (16.4)	16/45 (35.6)	
≥4	28 (15.1)	22/140 (15.7)	6/45 (13.3)	

2D, two-dimensional; TEE, transesophageal echocardiography; ICE, intracardiac echocardiography; RT 3D TEE, real-time three-dimensional transesophageal echocardiography; SD, standard deviation; ASD, atrial septal defect.

**Table 2 jcm-11-02394-t002:** Procedural data of the patients who underwent transcatheter device closure.

Variables	Total (%)	ICE (%)	3D TEE ± ICE (%)	*p-*Value
	*n* = 185 (100)	*n* = 140 (76)	*n* = 45 (24)	
Other features of ASD				0.8027
ASA	52 (28.1)	39 (21.1)	13 (7.0)	
Fenestrated	11 (5.9)	7 (3.8)	4 (2.2)	
Fenestrated with ASA	16 (8.6)	12 (6.5)	4 (2.2)	
Neither	106 (56.3)	82 (43.3)	24 (13.0)	
Distance between defects				<0.0001
<7 mm	120 (64.9)	103 (55.7)	17 (9.2)	
≥7 mm	62 (33.5)	34 (18.4)	28 (15.1)	
Difficult to measure	3 (1.6)	3 (1.6)	0 (0)	
Number of devices				<0.0001
1	146 (79.3)	129 (70.1)	17 (9.2)	
2	33 (17.9)	11 (6.0)	22 (11.9)	
3	5 (2.7)	0 (0)	5 (2.7)	
Under general anesthesia	52/185 (28.1)	7/140 (5.0)	45/45 (100.0)	<0.0001
Fluoroscopic time				
(min, mean ± SD)	17.0 ± 12.2	14.0 ± 6.2	24.9 ± 16.5	0.0005

2D, two-dimensional; TEE, transesophageal echocardiography; ICE, intracardiac echocardiography; RT 3D TEE, real-time three-dimensional transesophageal echocardiography; SD, standard deviation; ASD, atrial septal defect; ASA, atrial septal aneurysm.

**Table 3 jcm-11-02394-t003:** Outcomes and complications.

Variables	Total (%)	ICE (%)	3DTEE ± ICE (%)	*p-*Value
	*n* = 185 (100)	*n* = 140 (76)	*n* = 45 (24)	
Procedural success rate	182 (98.4)	139 (99.3)	43 (95.6)	0.147
Conversion to surgery	3 (0.2)	1 (0.7)	2 (4.4)	
Conversion to surgery	3 (0.2)	1 (0.7)	2 (4.4)	
Immediate outcomes				0.328
Complete closure	8 (4.4)	5 (3.6)	3 (6.7)	
Residual primary defects	169 (92.9)	129 (92.8)	40 (88.9)	
Other isolated defects	86 (47.2)	72 (51.8)	14 (31.1)	
Complication or mortality	3	0	3	
Procedural				
Device embolization	1	0	1	
Complete AV block	1	0	1	
Periprocedural		0	0	
Device embolization	1		1	
Long-term outcome				0.464
Complete closure	116 (63.7)	88 (63.3)	28 (62.2)	
Residual primary defects	35 (13.2)	23 (12.2)	12 (26.7)	
Other isolated defects	42 (23.1)	34 (24.5)	8 (17.8)	
Late complication (Device embolization, thromboembolism, complete AV block, cardiac tamponade, erosion)	0	0	0	
Duration of follow-up, months (mean ± SD)	49.7 ± 36.5	48.9 ± 35.8	51.9 ± 39.4	0.372

2D, two-dimensional; TEE, transesophageal echocardiography; ICE, intracardiac echocardiography; RT 3D TEE, real-time three-dimensional transesophageal echocardiography; AV, atrioventricular; SD, standard deviation.

## Data Availability

The data underlying this article will be shared at reasonable request to the corresponding authors.

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
