# Peer review of "Intracardiac Echocardiogram: Feasibility, Efficacy, and Safety for Guidance of Transcatheter Multiple Atrial Septal Defects Closure"

_jcm, 2022, doi:10.3390/jcm11092394_

Round 1

Reviewer 1 Report

This is a retrospective review of 2D ICE guidance, in comparison to the 3D TEE, to close multiple secundum ASDs regarding of the feasibility, efficacy, and safety. This review is worth to publish in order to provide additional information for this alternative imaging modality.  Nonetheless, some opinions the reviewer would like to share.

Title:

The main objective of this study is to focus on ICE guided multiple ASD closure, the reviewer would like to propose the optional title as “intracardiac echocardiogram: feasibility, efficacy, and safety for guidance of transcatheter multiple ASDs closure”

Materials and Methods:

Study protocol and population:

  1. How do the authors define “feasibility” and “safety” of using ICE?
  2. Please provide the definition of “multiple ASDs”.
  3. To the reviewer’s understanding, the authors categorize multiple ASD into the subgroup of “atrial septal aneurysm”, “fenestrated defect”, “fenestrated defect with atrial septal aneurysm”, and “neither”. Please consider to provide specific details of “neither” group.

Procedure technique

  1. Since there are 2 sizes of AcuNav system, 8F and 10F, please provide the detail of the AcuNav the authors used.

Results

Row 156: please consider to use abbreviation “ICE” for “intracardiac echocardiogram”.

All the tables: please consider to describe number after decimal in the similar pattern throughout the tables.

Table1 and 2: for abbreviation legend “RT 3D TEE”, please consider to modify this abbreviation to “3D TEE” to match with what mentioned throughout the manuscript.

Outcomes and complications:

For procedural complication, ICE can cause vascular injury and cardiac arrhythmia (especially in small children), please consider to include them in the manuscript. In addition, please consider to provide more details of patients with BW < 10 kg who received ICE guided ASD closure.

Discussion

  1. As the authors stated in the objective of this study, to assess the feasibility and safety of ICE to guide multiple ASD closure, please consider to discuss on feasibility and safety of using this imaging technique (especially in children).

Figure 4 and 5: there are no reference of these 2 figures in the manuscript, please consider to provide further discussions.

Figure 5:

  1. Please consider to adjust abbreviation of “RT3D” and “RT3D TEE” to “3D TEE” as described in the manuscript.
  2. Figure legend “RT3D TTE”, please consider to modify it to “3DTEE”

Conclusion:

  1. The author stated that “In some selected multiple ASDs, ICE could be a safer and more efficient………”. What Does “some selected multiple ASDs” mean, ASDs with a minor distance between 2 significant defects and those in whom one device was expected to close another defect?
  2. The last sentence of conclusion, the authors mentioned about the new technology, EchoNavigator. How do the authors think about 3D ICE?

Author Response

We appreciate your acknowledgement of our study. Your comments have encouraged us to improve this study and add more valuable information. Thank you very much.

For "point by point response to the reviewer's commnets", please see the attachement. 

Reviewer 2 Report

Major comments:

This is a quite interesting manuscript of a single institution experience to discuss the feasibility, accuracy, effectiveness and safety of intracardiac echocardiography (ICE) for closing multiple atrial septal defects in comparison to those of 3D transesophageal echocardiography. (3DTEE)

As the authors described, ICE assessment for multiple ASDs in younger age with <30Kg is very beneficial in clinical setting, if those assessment were almost same as those of 3D-TEE, because 3D-TEE has the difficulties to perform the children <30Kg.

A key of this manuscript is how ICE can be comparable to 3D-TEE to provide a good and sufficient quality of imaging for closing multiple ASDs in the cath lab.

In this regards, the authors provided good enough number of patients and clinical data for their conclusion.

The style and logic were consistent and well described with comprehensive English.

  1. The reviewer could understand the success rate and the complication event rate between by ICE and by 3D-TEE when to close multiple ASDs.However, there was few description about the agreement of the anatomical findings and measurements between ICE and 3D-TEE, such as defect size, rim length, inter-defect distance, the shape of defects and so on. These measurement should be very important for making the decision of intervention and device size selection. It would be preferrable to mention about the agreement and disagreement of between ICE and 3D-TEE findings in this study. If any disagreements, it would be better for the authors to explain why and how.
  1. The 3D morphological imaging of multiple ASDs is essential as the authors described. The 3D-TEE is widely recognized of its superiority and usefulness to understand 3D morphology. Instead, ICE provides only 2D images. How did the authors reconstruct the 3D morphological structure from ICE images? If any tips, please describe.
  2. Are there any difference between ICE and 3D-TEE to assess images before and after the device closures?
  3. This report from the high volume center using ICE for closing ASDs. Does this fact affect the authors' conclusion of this article?

Author Response

We appreciate your systematic review of our study. Your comments have encouraged us to improve this study and add more valuable information. Thank you very much.

For " point-by-point response to the reviewer's commnets", please see the attachment. 

Reviewer 3 Report

The manuscript determined the feasibility, efficacy, success, and safety of 2D ICE and 3D TEE in  transcatheter ASDs closure. Even though it is retrospective study, the subject of the paper is of potential interest, there are a number of major limitations of the manuscript:

  1. In the "Abstract" section the aim of the study is "We aimed to determine the feasibility, efficacy, success, and safety of intracardiac echocardiography (ICE) and three-dimensional (3D) transesophageal echocardiography (TEE) in transcatheter multiple atrial septal defects (ASDs) closure" while in the introduction section is " Here, we wanted to assess through retrospective observation the feasibility, efficacy, procedural success, and safety of ICE in transcatheter multiple ASDs closure." Please clarify what is the aim of the study.  
  1. The authors should state more clearly according to their result in which situations we should use 3D TEE and when 2D ICE.
  2. Some of the results are simply confirmatory and the clinical implications are not clearly addressed. What is the meaning of their results? How would they change the clinical practice?
  3. The manuscript has important concerns/drawbacks, some of them assumed by the authors in the section limitations. However, this assumption does not make the results of the research valid. So, the conclusions of the study must should be more concise.

Author Response

We appreciate your acknowledgement of our study. Your comments have encouraged us to improve this study and add more valuable information.

For the "point-by-point response to the reviewer’s comments", please see the attachment.

Thank you very much.

Round 2

Reviewer 3 Report

I have read the latest revision of your manuscript, entitled "Intracardiac Echocardiogram: Feasibility, Efficacy, and Safety for guidance of Transcatheter multiple Atrial Septal Defects closure?". Thank you for addressing the issues that we asked you to consider in an effective manner, and for indicating changes clearly in your cover letter and in the revised text. I do not have further comments. 

Author Response

Dear JCM reviewer
We sincerely thank you for the positive evaluation of our revision. Tody we upload the final final version. There are no more changes in content.